# A Reasoning-Based Approach to Cryptic Crossword Clue Solving

**Martin Andrews** [1]   **Sam Witteveen** [1]

## Abstract

Cryptic crossword clues are challenging language tasks for which new test sets are released daily by major newspapers on a global basis. Each cryptic clue contains both the definition of the answer to be placed in the crossword grid (in common with regular crosswords), and 'wordplay' that *proves* that the answer is correct (i.e. a human solver can be confident that an answer is correct without needing crossing words as confirmation). This work describes an LLM-based reasoning system built from open-licensed components that solves cryptic clues by (i) hypothesising answers; (ii) proposing wordplay explanations; and (iii) using a verifier system that operates on codified reasoning steps. Overall, this system establishes a new state-of-the-art performance on the challenging Cryptonite dataset of clues from The Times and The Telegraph newspapers in the UK. Because each proved solution is expressed in Python, interpretable wordplay reasoning for proven answers is available for inspection.

## 1. Introduction

There has been significant work in reasoning in the fields of mathematics (Jiang et al., 2023; Yang et al., 2023; Trinh et al., 2024) and code generation (Ni et al., 2023; Ridnik et al., 2024) which benefit from having strong verifiers to validate their answers. This work tackles the relatively under-studied reasoning task of cryptic crossword solving, which has the following qualities:

- Thousands of people find Cryptic Crosswords a satisfying intellectual challenge on a daily basis. Solving these puzzles requires understanding multi-layered language constructs, blending logic, wordplay, and contextual nuance. This provides a unique challenge for evaluating and improving LLMs' capabilities in NLU and reasoning.

[1]Red Dragon AI, Singapore. Correspondence to: Martin Andrews <martin@reddragon.ai>.

*Proceedings of the 42nd International Conference on Machine Learning*, Vancouver, Canada. PMLR 267, 2025. Copyright 2025 by the author(s).

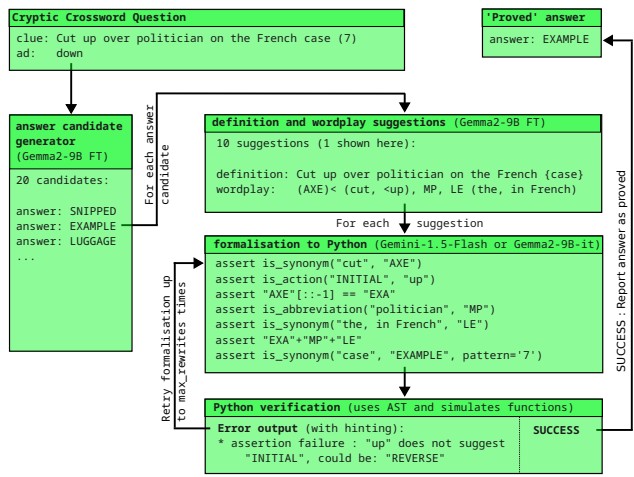

*Figure 1.* Proving process: `answer` candidate → `wordplay` → LLM formalisation

- There are decades of solved puzzles (each one containing over 20 clues) from multiple major newspapers available, and new puzzles are published daily. This contrasts with (for instance) IMO/AIME problems, where there is a much lower number of novel problems available.

- The method in this work explicitly reveals the reasoning (i.e. validated wordplay) required to solve each problem. By construction, there is one 'true' reasoning path, although it might be expressed in different ways by different solvers.

### 1.1. An example Cryptic Clue

As a concrete example, and to clarify the terminology used, consider the following *moderately* complex cryptic clue[1] : "Cut up over politician on the French case (7)".

Solvers must parse the clue carefully to separate the `definition` (which acts as a regular crossword clue) and the supporting `wordplay` that can be used to arrive at the same `answer` from two directions. Arriving at the same answer by two paths constitutes the necessary proof that the correct answer has been found. See Figure 2a for a visual depiction of the reasoning involved.

---

[1]The Times UK (6-March-2024), Cryptic #28857 clue 4D

In this work, we take our cue from the effectiveness of provers coupled with verifiers for mathematical reasoning tasks (Jiang et al., 2023). We tackle the cryptic crossword clue solving task using an LLM to (i) suggest `answer` candidates; (ii) create informal proofs (i.e. coming up with `wordplay`); and (iii) perform a formalisation process (which rewrites the `wordplay` logic in Python). The proposed solutions (expressed as executable Python) are then checked for validity.

### 1.2. Contributions

The following are the main contributions of this work:

- **An open-license system for reasoning over Cryptic clues** - our pipeline (illustrated in Figure 1) enables 9B-scale local models to achieve state-of-the-art results on the Cryptonite dataset.

- **Local models for cryptic clue tasks** - We show how local LLMs can be fine-tuned to produce `answer` candidates, and `wordplay` suggestions, and then prompted to perform Wordplay formalisation. Following an approach akin to mathematical statement formalisation, but where there are *less than 10* examples of 'good proofs' available, our novel pipeline was specifically engineered to avoid 'reasoning steps' becoming stuck in dead ends.

- **Python domain-specific verifier** - Using the output of the formaliser, the verifier presented here deconstructs the Python AST, so that it can evaluate each `assert` statement on a line-by-line basis. We believe that this is somewhat novel, since it enables the verifier to not only indicate whether the proof is valid overall, but also point to specific failures (used to regenerate failed formalisations) on all proof lines simultaneously.

To promote further study in this area, all code for training the models, the formaliser and domain-specific verifier is made publicly available.

## 2. Related Work

### 2.1. Regular Crosswords

Non-cryptic ("regular") crosswords are known throughout the world, and are the predominant type found in newspapers in the U.S.A. One key difference from cryptic crosswords is that individual regular crossword clues are generally not 'standalone' - there may be a number of different answers that fit the given clue. The key to solving regular crosswords is thus the interaction between answers (i.e. the crossing-words), which allows for planning/backtracking to enable solving rates in the high 90% range (Wallace et al., 2022).

This work, in contrast, focuses on the solving of clues on a standalone basis, which requires elements of reasoning through the wordplay present in cryptic clues.

### 2.2. Cryptic Crosswords

In an 800 participant research study into the backgrounds of cryptic crossword solvers (Friedlander & Fine, 2016), the following observation was made about the skills required to solve these linguistic/reasoning puzzles:

> "... cryptic crossword skill therefore appears to be bound up with code-cracking and problem-solving skills of a quasi-algebraic nature. Conversely, lexical ability, although no doubt valuable, does not appear to be a critical discriminator of high expertise among elite solvers."

Cryptic crosswords have received surprisingly little attention from the machine learning community, despite being a notable language-oriented reasoning puzzle with global appeal. One possible reason is that cryptic crosswords are much less common in the United States than 'regular crosswords'. See Anthony & Goodliffe (2024) and Webb (2024) for inspiring demonstrations of experts solving cryptic crosswords in real-time.

The benchmark dataset used by this work is Cryptonite (Efrat et al., 2021) - a large-scale dataset of Cryptic Crossword clues from The Times and The Telegraph (major UK newspapers). The dataset contains 523,000 naturally sourced clues (published between 2001 and 2020), with the train, validation and testing splits being chosen so that a given `answer` can only appear in one of the splits.

While the dataset made available in Rozner et al. (2021) is also of interest, its clues are limited to those from the Guardian newspaper, and Connor (2024) notes in the Guardian's own blog "The Times hosts an annual crossword-solving competition and it remains, until such time as the Guardian has its own version, the gold standard." Moreover, the smaller number (142,000) of clues that the dataset contains have no orientation markings ('across/down'), which are required to make sense of some `wordplay`.

For a more in-depth discussion of the decision to focus on the Cryptonite dataset (and not perform testing on the Guardian dataset), please see Appendix A.3. In summary, while the 'Init' split presented in Rozner et al. (2021) has attractive properties (explored there, and in other works), this work specifically targets the reasoning side of cryptic clues, which involves fine-tuning models on Cryptonite (including Wordplay examples with carefully matched train/val/test splits). This precludes us from doing the same kind of multi-dataset comparisons found elsewhere.

#### 2.2.1. RULE-BASED SOLVERS

Williams & Woodhead (1979) is an early example of attempting to devise a formal language for describing cryptic clues. However, the linguistic elements of the clues tend to

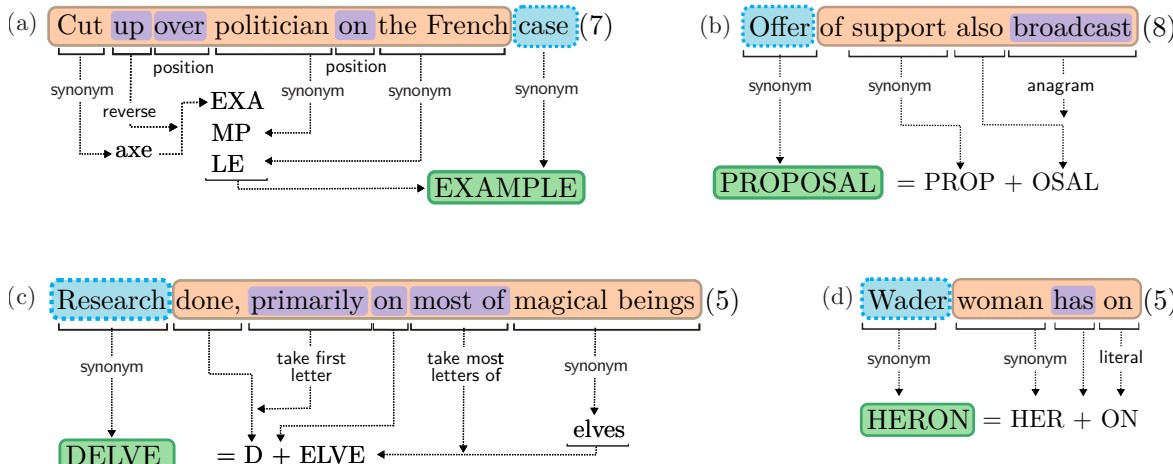

Figure 2. Clue solving illustrations. Answers are in green, definitions in blue (dashed frame), wordplays in orange, and indicators in purple. Further textual examples can be found in Appendix A.1

thwart a strictly formal approach.

A more flexible rule-based solver with a manually-crafted probabilistic grammar was introduced in Deits (2015; 2022). Building on the assumption that a `clue` can usually be split into `wordplay` and `definition`, the solver tries to find the most probable parse such that the `wordplay` yields a semantically-similar result to the `definition`. The logical form of this DSL approach is very appealing. However, it appears limited to solving clues where the `wordplay` is somewhat simple (due to the combinatorial explosion of possibilities created by longer/more complex clues).

The goal of this work is to use the flexibility of LLMs to enable a far wider range of clues to be attempted, with the aid of a formaliser/verifier to check the solutions.

### 2.2.2. LLM-BASED SOLVERS

Cryptonite is a challenging task for LLMs : Efrat et al. (2021) reported that fine-tuning T5-Large (a 770M encoder-decoder model) on Cryptonite's 470k cryptic clue training set achieved only 7.6% test set accuracy, slightly below the 8.6% accuracy of the rule-based clue solver of Deits (2022). Interestingly, prior to 2024, even large-scale Language Models scored very poorly on cryptic clues, likely due to (i) the misleading surface reading of the clues; (ii) the obliqueness of the definitions; and (iii) the reasoning steps required to *prove* the answer correct based on the wordplay.

Recent works, such as Sadallah et al. (2024) and Saha et al. (2024), tackle cryptic crosswords with more up-to-date local models and commercial LLMs. Saha et al. (2024) reports results with 5- and 10-Shot prompting (without fine-tuning the models), but also includes a wide-ranging study of the capabilities of models for crosswords in general. We include experiments that bring the relevant baselines up-to-date, and

also touch on their illuminating Partial Correctness Metrics (which are relevant when attempting full grids, which is not the main focus here).

In this work, building on the groundwork of Andrews & Witteveen (2024) and Andrews & Witteveen (2025), we use a pipeline of 9B-scale LMs to produce `answer` candidates and `wordplay` suggestions, followed by a third LM to formalise each proposed solution using Python code and then rewrite/update the solutions based on feedback from a purpose-built verifier. In our results, we focus on the 'pure' Cryptonite benchmark: Accuracy is judged based on a Top-1 basis (with the model's single `answer` being marked wholly correct or not), with no crossing letters being given. Framed as a reasoning task, if the model 'understands' the cryptic solution properly, the `answer` will be wholly correct - there should be no partial marks.

### 2.3. Code & reasoning

To compensate for LLM *approximate generation* of logical reasoning, techniques like PAL (Gao et al., 2023) exploit LLMs' facility for writing code to create verifiable reasoning chains. An important influence on this work was also the Draft, Sketch, and Prove framework (Jiang et al., 2023) which uses an LLM to draft and create proofs that are then verified formally.

As with the prior LM autoformalisation work Ye et al. (2023), we chose to use Python as the intermediate language into which the natural language statement was formalised. In our case, rather than using an external prover, our system formalises its proofs directly in Python, using callable functions, such as `is_anagram()`. This light-DSL approach was essential for our NLP task, since we found (for instance) that LLMs have trouble recognising whether two sequences

```
"publication": "FT",
"setter": "falcon",  "author": "teacow",
"num": 16, "ad": "D", "pattern": "8",
"clue": "{Offer} of support also broadcast",
"wordplay": "PROP (support) +
            (ALSO)* (*broadcast)",
"answer": "PROPOSAL"
```

*Figure 3.* An example from the Wordplay dataset (in this wordplay, ()* is an anagram indicator). This clue's solution is diagrammed in Figure 2b

of letters are anagrams of each other.

In contrast with the tool-integrated reasoning framework Gou et al. (2024), where an LLM for mathematical problem-solving was fine-tuned on 16,000 examples of formalisation, we found that our light-DSL was able to be used by LLMs based on its in-context description alone. For full prompting details, please see Appendix A.5.4.

Informed by the evolution from AlphaCode (Li et al., 2022), in which huge numbers of programs are generated and filtered in order to generate a valid solution, to AlphaCodium (Ridnik et al., 2024), in which solutions are iterated upon and involving much less computation, this work uses a verifier that can feed back 'hints' to the formalising LLM, so that the task of re-writing nearly-valid proofs is made easier.

### 2.4. Wordplay dataset

The Wordplay dataset (Andrews, 2024) - an example from which is given in Figure 3 - consists of data gathered from websites where cryptic crossword enthusiasts post solutions on a daily basis for each of the major publications. Each completed puzzle is annotated by an individual, identifiable author/solver that lists the approximately 20 clues with their definition, wordplay and answer fields. Note that each solver can chose their own 'standard' for writing out the wordplay, leading to a significant variation in wordplay annotation styles (even across time for an individual solver). The Wordplay dataset deliberately follows the train, validation, and test splits defined by Cryptonite.

## 3. Methods

The overall system described in the work is illustrated in Figure 1, and the code[2] is available under an Apache 2 license. The order of operations for the pipeline was chosen based on watching human solvers - who report going through the following steps: (a) attempting to parse the clue in a number of ways, trying to isolate the definition from the wordplay; (b) seeing which parts of the wordplay they

are most confident about; (c) 'having a hunch' of the final answer; and (d) gaining a full understanding of how a clue's wordplay works (such that the function of every element can be explained) as proof of the overall process.

Concretely, the system starts with the clue, and generates 20 answer candidates. For each unique candidate, the next step is to generate 10 guesses at wordplay to justify the answer. Then, each of these wordplay hypotheses are 'formalised' as a Python program in a particular form, which can then be verified by executing it (with several retries in case of failure). Successful executions are taken as *proof* that the original answer was correct.

Observations of the behaviour of GPT-4 using Chain-of-Thought prompts (Wei et al., 2022) suggest that even very capable models tend to fixate early on during the reasoning process, and are only rarely able of completely re-hypothesising. These LLMs also frequently becomes caught up with the literal 'surface' meaning of the clue, which is often misleading (deliberately on the part of the setter). Organising our system's pipeline to hypothesise candidate answers as the first step (so that the models must try to fit the reasoning to the answer, with varying degrees of success) bakes re-hypothesisation into the process.

### 3.1. Candidate answer generation

Our first step to solving a given cryptic clue is to generate multiple answer candidates from the original clue, pattern and ad (across/down) fields. For this task, we fine-tuned a Gemma2 9B base model (Gemma Team & Google DeepMind, 2024) using the LoRA (Hu et al., 2022) implementation provided by the unsloth package (unsloth.ai, 2024). The model was trained for 1 epoch on the Cryptonite training set of approximately 470,000 examples.

For each clue being evaluated, we generate 20 valid answer candidates, where candidates that did not match the pattern were immediately rejected and regenerated, and those not contained in the crossword words list (Beresford, 2000) were filtered out[3]. The number of candidates was chosen to balance generation cost with likelihood of the correct answer appearing in the candidate list - see Figure 7 for a cumulative frequency analysis. The list of candidates was then grouped so that the frequency of each answer could be found - enabling statistics to be collected.

### 3.2. Generation of definition and wordplay suggestions

To train the wordplay suggestion model, which translates each answer candidate into multiple definition

---

[2]https://github.com/mdda/cryptic-crossword-reasoning-verifier

---

[3]This rejection of invalid words here is not 'cheating' since we do not use the dictionary to suggest words, rather it is only used to weed out actively proposed non-words from a short-list.

```
def proof(answer="DELVE",
          clue="research done, primarily on most of magical beings",
          pattern='5'):
```
> **Gemma2 answer candidate**

```
    """
    definition: research done, primarily, on most of magical beings
    wordplay: D[one] (primarily) ELVE[s] (magical beings, most of)
    """
```
> **Gemma2 wordplay**

```
    assert action_type("primarily", Action.INITIALS)
    assert "DONE"[:1] == "D"
    assert is_synonym("magical beings", "ELVES")
    assert action_type("most of", Action.REMOVE_LAST)
    assert "ELVES"[:-1] == "ELVE"
    assert "D"+"ELVE" == "DELVE"
    assert is_synonym("research", "DELVE", pattern='5')
proof()
```
> **LLM formalisation**

*Figure 4.* Python proving: `answer` candidate → `wordplay` → LLM formalisation

and `wordplay` suggestions, we make use of the Wordplay dataset of Andrews (2024). For this task, we fine-tuned another Gemma2 9B base model using LoRA. The model was trained on 4 epochs on a set of approximately 16,800 examples (consisting of solution explanations of puzzles from The Times and The Financial Times from selected authors in the Wordplay dataset).

### 3.3. Python formalisation

Rather than create a dataset with many examples of formalisation, here we use in-context prompting with less than 10 examples of the formalisation style required. In preliminary work, we concluded that the available Gemini-Flash LLM was not capable of using a (novel) cryptic crossword domain specific language ("DSL") through in-context learning with so few examples. In contrast, we found that the LLM could be prompted to produce Python code with relative ease, so the approach taken was to frame a declarative-style-DSL as Python function calls within `assert` statements. The LLM was found to be able to reliably produce syntactically correct Python, and use the 'external functions' that had been described (as illustrated in Figure 5) to form logical sequences of declarations, which could then be parsed line-by-line by manipulating the Python abstract syntax tree ("AST"). An example of the Python DSL being generated by the formalisation LLM is given in Figure 4, with the workings of the clue solution being illustrated in Figure 2c.

To formalise `wordplay` into Python 'proofs' of the correctness of solutions, we used Google's `Gemini-Flash-1.5-001` LLM (a pinned model version) during development. This model was initially chosen instead of a frontier-tier model since the formalisation task should not require much inventiveness/reasoning: the actual required steps are already present in the `wordplay`, the task is *merely* to translate to Python. To determine whether the choice of `Gemini-Flash` was

```
Action=Enum('Action',
            'ANAGRAM,REMOVE_FIRST,INITIALS,REMOVE_LAST,'+
            'GOES_INSIDE,GOES_OUTSIDE,REVERSE,SUBSTRING,HOMOPHONE')
# External definitions
def is_synonym(phrase:str, test_synonym:str, pattern:str='') -> bool:
  # True if 'test_synonym' is a reasonable synonym for 'phrase',
  # with letters optionally matching 'pattern'
def is_abbreviation(phrase:str, test_abbreviation:str) -> bool:
  # Determines whether 'test_abbreviation' is
  # a valid abbreviation or short form for 'phrase'
def action_type(phrase:str, action:Action) -> bool:
  # Determines whether 'phrase' might signify the given 'action'
def is_anagram(letters:str, word:str) -> bool:
  # True if 'word' can be formed from 'letters' (i.e. an anagram)
def is_homophone(phrase:str, test_homophone:str) -> bool:
  # Determines whether 'test_homophone' sounds like 'phrase'
```

*Figure 5.* External functions available via In-Context Learning

a limiting factor, we subsequently tested an unmodified `Gemma2-9B-it` model on the same task.

In terms of the DSL itself, the back-end to the `is_synonym` and `is_homophone` functions consists of calls to simple language models. The `action_type` function performs a nearest-neighbour match against list of indicator words, and the `is_abbreviation` function performs a look-up against a list of abbreviations - both sourced from Deits (2022). For string manipulation actions (such as 'REVERSE'), the LLM formaliser itself was capable of producing correct string manipulation expressions unaided.

### 3.4. In-Context Learning

To produce Python code that could be sent to the prover, the LLM was prompted in an In-Context Learning ("ICL") manner. This consisted of the following parts:

1. Cryptic crossword rubric to explain to the LLM what the principles were behind the fields such as `clue`, `definition`, `wordplay`, etc.

2. 20-shot examples `clue` → `wordplay`

3. The 'external functions' rubric shown in Figure 5

4. Few-shot `wordplay` → `Python` formalisations (6 examples given)

5. The actual `clue`, `answer`, `definition` and `wordplay` being formalised

Gemini-Flash did not appear to be particularly sensitive to the prompting style used, except in the 'handover step' (between problem description and model generation) where several trials were needed to obtain the final function definition in the required format consistently. Further details of all the ICL prompts are given in Appendix A.5. For the Gemma2-9B-it formalisation runs, the same prompts were used unchanged (with no other tuning/training). In additional, a further Gemma2-9B model was trained on 448 *valid* Gemini-created proofs of ground-truth Wordplay examples.

```
AssertionError: is_abbreviation('an Artist', 'RA'):
   'an Artist' does not have a valid abbreviation;
   'RA' is an abbreviation for :
     artist, artillery, Royal Artillery, gunners, painter
AssertionError: action_type('goes crazy', Action.ANAGRAM):
  'goes crazy' does not suggest Action.ANAGRAM,
  but 'crazy' does
AssertionError: action_type('worked', Action.HOMOPHONE):
  'worked' does not suggest Action.HOMOPHONE,
  but may be Action.ANAGRAM
```

*Figure 6.* Illustrative `AssertionError` responses (with hinting) from the verifier

*Figure 7.* Statistics of `answer` candidate list, as more candidates generated

## 4. Experiments

### 4.1. Gemma2 9B `answer` **candidate generation**

During the initial experimental phases of fine-tuning local models for the `answer` generation task it was discovered that `-base` models scored more highly than `-it` models. This might be explained by observing that instruction fine-tuning may (to some extent) penalise off-the-wall answers, which may be essential for our task. In addition, we also observed that while the Top-1 candidate from a model generating with a temperature $t = 0.5$ had high accuracy, it was beneficial to run candidate generation with $t = 1.0$ (even though the Top-1 accuracy was lower in this case) - since having a wider spread of `answer` candidates was useful for our pipeline overall.

Figure 7a shows that the probability of the gold answer being among the candidates produced is (unsurprisingly) monotonically increasing in the number of independent samples. It also shows that this process is not yet asymptotically limited, although slowing down with increasing $n$.

Figure 7b shows that choosing the highest-frequency `answer` candidate can be a very effective strategy. However, there is a clear limit to this idea: There is a significant probability that cryptic crossword answers are in the long tail of potential answers. Indeed, intentionally creating misleading clue 'surface readings' is a hallmark of good cryptic clue setting.

### 4.2. Gemma2 9B `wordplay` **candidate generation**

Since `wordplay` is so flexible, it is difficult to evaluate it for accuracy against other examples (without, say, a large LLM to evaluate the differences). However, good `wordplay` should result in good formalisations, so evaluation is available on an end-to-end basis.

One key assumption in the system proposed here is that a correct `answer` should lead to interpretable `wordplay`, whereas an incorrect `answer` candidate should give rise to unformalisable/unverifiable `wordplay`. The following typical example illustrates how the correct `answer` leads readily to correct `wordplay` (the workings of this clue

### 3.5. Proof Verification with Hinting

The system's verifier must decide whether a given formalisation is valid, and report any errors found to iteratively improve the Python code as feedback to the LLM formaliser in a cycle, as seen in Self-Debug (Chen et al., 2024), and AlphaCodium (Ridnik et al., 2024). Examples of assertion failures, with constructive hinting, are shown in Figure 6.

This cycle is repeated until a formalisation is validated (zero assertion failures, considered a 'SUCCESS' with the `answer` having been proved), or `max_rewrites=2` is reached. If no Python formalisation can be validated, then the fallback `answer` is used (defined as being the most frequent `answer` amongst the original candidates produced in the first stage of solving the `clue`).

### 3.6. Partial Correctness Metrics

One interesting direction explored in Saha et al. (2024) was the performance of LLMs on cryptic clues if some of the letters were known (as would be the case if an entire grid were being solved). The conditions examined were with 25%, 50% and 70% of letters 'known'. Based on observations working with the proposed system for single clues (and more general experience of crossword solving), we approached this problem in two ways.

For the 25% level of letters 'known', it was a simple matter to use the current system with candidate answers which didn't match the known letters filtered out. For the higher levels of known letters, we instead used the FastText embedding method of Mikolov et al. (2018) to find the nearest neighbour `answer` within The UK Advanced Cryptics Dictionary, Beresford (2000), by comparing against the embedding of the raw `clue` phrase itself.

The 'Partial Correctness' results, since they are not the core thrust of this work but interesting in their own right, are given in Appendix A.5.7.

are illustrated in Figure 2d), whereas trials with an incorrect `answer` candidate (which was, in fact, the most frequent candidate for this `clue`) give clearly unverifiable `wordplay`:

```
clue: "wader woman has on (5)"
    definition: "{wader} woman has on"

  answer: "HERON"  # correct answer
    wordplay: "woman (HER) has on (ON)"

  # incorrect answer - 3 trials shown
  answer: EGRET
    wordplay: "woman (HER) on top of
      (REG - another word for on, as in
      'do you have the heating on?')"
    wordplay: "EG (woman has) + RET (on)"
    wordplay: "woman (HER) has on/around
              (EG) - a wader bird"
```

### 4.3. Cryptonite Results (Top-1 exact match)

In this work, we focus our testing on using the Cryptonite dataset of Efrat et al. (2021) as a benchmark, with the Top-1 exact-match results shown in Table 1. As in Saha et al. (2024), due to computational constraints, we performed sampling of the validation and test sets, using fewer than the full 26k examples available. The standard deviation of these figures is $\approx \pm 1.5\%$ at 1000 samples, and $\approx \pm 3.3\%$ at 200. To determine whether the systems presented here 'beat' GPT-4o, we performed a Bayesian Item Response Theory test (Fox, 2010) to estimate the probability that our results outperformed the GPT-4o (over the same samples).

The 5-Shot results in Table 1 show that:

- GPT-4o (2024-11-25) gives stronger results than those of GPT-4-Turbo (2024-04-09) given in Saha et al. (2024) - so this is an updated baseline;

- The updated GPT-4o results show surprisingly strong performance on the validation split (unfortunately, the composition of this commercial model's training data is unknown);

- Gemini-1.5-Flash-001 (which was used in development of the formaliser) is not particularly good at solving cryptic clues in itself;

- The Gemma2-9B model gets a large uplift from fine-tuning on the Cryptonite training set (compare the 5-Shot figures to the later Gemma2-9B FT ones).

The `Gemma2-9B FT` accuracy figures are for the first result returned by the fine-tuned Gemma2 model. In contrast, the `Gemma2-9B freq` accuracy figures are for the most common (i.e. highest frequency) result among the Gemma2 `answer` candidates (for which 20 samples were generated for each `clue`). These voting-based results would have exceeded prior state-of-the-art results for open-licensed models on their own.

Going beyond single models, the `Gemini-Flash Formaliser` demonstrates Top-1 exact-match performance of 32.5% for the Cryptonite Test set, establishing a new state-of-the art result against the updated baselines (the Bayesian IRT results are that Gemini-Flash has a probability of 92% of being actually better than GPT-4o).

Moreover, the results of the non-fine-tuned `Gemma2-9B-it Formaliser` also (marginally) beat the previous state-of-the-art results - which is perhaps an even stronger statement about the capabilities the system described here, since in this case Gemma2-9B models have been used throughout the solving process, showing that it is be possible to achieve very competitive cryptic crossword solving results through reasoning with open-licensed models. The Bayesian IRT results are that the Gemma-9B FT model has a probability of 81% of being actually better than GPT-4o on Hard clues, 57% overall.

The formaliser results are (surprisingly) relatively worse for Quick clues. This seems to be related to the fact that the agreement/frequency-based `Gemma2 freq` model is very strong on these clues, and any 'contribution' from the formalising/verification procedure is likely to overrule a good base-line result, due to erroneous verification of 'proofs' that are not valid.

### 4.4. Ablations

The lines in Table 1 marked '(AB)' are ablations. Both utilise the measurement of average *logprob* of the output tokens given by the relevant model.

The first ('logprob answer') shows the results of using the candidate answer generation Gemma2-9B FT model from above, with the candidate `answer` being chosen from the list of 20 possibilities according to highest *logprob*. Since answers are typically very short, this method is similar to the frequency-based selection model.

The second ('logprob wordplay') shows the results of evaluating the Gemma2-9B FT model that generates `wordplay` hypotheses, and choosing an `answer` based on the highest *logprob* according that generating model. Somewhat unexpectedly, this was not as effective as might be assumed from the generated `wordplay` seen in Section 4.2 - where the `wordplay` for wrong answers looks absurd. Examining samples of the `wordplay` most favoured by pure *logprob* order, it seems that the generating LLM finds simply-worded but *completely fictitious* `wordplay` quite likely.

Both of these ablations demonstrate that the formalisation and verification steps are essential components in our system

*Table 1.* Cryptonite results : Standard splits, Top-1 answer accuracy rate

| Model | samples | Validation | | | Test | | |
|---|---|---|---|---|---|---|---|
| | | Overall | Quick | Hard | Overall | Quick | Hard |
| Rule-based (*) | 26k | 8.3% | | | 8.6% | 13.5% | 5.8% |
| T5-large (770M) FT (*) | 26k | 7.4% | | | 7.6% | 12.8% | 3.4% |
| Gemma2-9B-it 5-shot | 1000 | 5.7% | 11.5% | 5.2% | 4.5% | 10.5% | 4.0% |
| Gemini-Flash 5-shot | 1000 | 6.6% | 12.5% | 6.1% | 6.5% | 11.8% | 6.1% |
| GPT-4o 5-shot | 1000 | **29.8%** | **45.0%** | **28.5%** | 27.6% | 47.4% | 26.0% |
| Gemma2-9B FT | 1000 | 21.7% | 28.8% | 21.1% | 15.9% | 38.2% | 14.1% |
| Gemma2-9B freq (#=20) | 1000 | 26.6% | 31.3% | 26.2% | 25.5% | **55.3%** | 23.1% |
| (AB) logprob answer | 500 | 23.9% | 35.9% | 22.9% | 22.7% | 55.3% | 20.1% |
| (AB) logprob wordplay | 200 | 21.0% | 15.4% | 21.4% | 20.5% | 46.7% | 18.4% |
| Gemini-Flash Formaliser | 200 | 28.0% | 23.1% | 28.3% | **32.5%** | 46.7% | **31.4%** |
| Gemma2 9B-it Formaliser | 200 | 26.0% | 23.1% | 26.2% | 29.0% | 46.7% | 27.6% |
| Gemma2 9B-FT Formaliser | 200 | 27.0% | 23.1% | 27.3% | 29.5% | 53.3% | 27.6% |

Rows (*) are as reported in Efrat et al. (2021); The Hard columns are for the non-Quick clues

- they cannot be shortcut by a 'dumb ranker' in the pipeline.

## 4.5. Qualitative Error Analysis

In addition to the numerical results presented in Table 1, we note the following qualitative aspects of our system's performance:

- The headline success rate is bounded above by the initial candidate answer generation process. If the system cannot guess the answer in its top-k ($k = 20$ here), the remaining process is doomed. As shown in Figure 7a, even with higher top-k, this puts an upper bound on performance that is well below 100% correct. Having better candidate answer generation would be beneficial - and this would directly feed through our verification process

- While the proprietary models may output the correct final answer, it is often the case that their 'reasoning process' makes no logical sense (indicating, perhaps, that they have memorised `clue`/`answer` pairs). In contrast, our method *does* give us useful human-interpretable reasoning for each solution

- A significant source of false negatives is the `is_synonym` function, which relies on a sequence of steps: first we attempt a look-up in an open-source thesaurus, then in a dataset of 'regular crossword answers'. But the final fall-back is asking an LLM whether given phrases are synonyms. While the first two steps may vote positively (for easy matches), it is common in cryptic clues that the definition and the answer are more distantly related than regular crosswords. For instance, in

Appendix A.1.7, we have the true answer `UNDERMINED` being defined by 'damaged'. This would likely be too distant to be reasonable for a regular crossword, but the strength of the `wordplay` (the answer being literally given in the clue) is confirmation enough to satisfy solvers. Setting this 'synonym distance hurdle' is an ongoing challenge.

## 4.6. Known Limitations of the System

While the verifier implemented for this work is effective, it does not completely cover the following possible potential 'shortcuts' in the Python functions it analyses:

- The entire Python function might consist of comments, so that nothing could trigger an `assert`. This has been partly countered by requiring the Python code to include at least 2 `assert` statements

- The Python function contains conditional execution, routing around `assert` statements

- Occasionally, the hint `assert XYZ failed` results in the re-write : `assert XYZ==False`, which is clearly not productive

- The proof may be logically disconnected, with left-hand-side terms not being supported / justified by right-hand-side terms in other lines of the code

These issues do not appear insurmountable, given time and effort. It should be noted that since the formalising LLM

is only being used In-Context there is little chance that the above issues are being systematically abused (which would almost certainly happen if there was learning-in-the-loop in a Reinforcement Learning setting).

## 5. Conclusions

The authors recognize the domain-specificity of cryptic crossword solving. However, we believe that it serves as a rigorous and complex test-bed for reasoning, requiring multi-faceted language understanding and logic. While the specific DSL developed here is tailored to crosswords, the underlying principles of our approach – decomposition, formalization, and verification with feedback – are intended to be more broadly applicable to other reasoning tasks. Cryptic crosswords, with their clearly defined rules and solutions, allow for precise evaluation and iterative refinement of these principles.

Our results have validated our overall approach to codification of the cryptic crossword problem domain : Generating `answer` candidates and `wordplay` suggestions followed by production of code via an LLM-based formalisation process, verification using Python code analysis tools, and iterative prompting of the LLM formaliser proved quite effective. Generating multiple candidate answers, followed by multiple wordplay samples, can been framed as inference-time computation (using 9B models) rather than using a large proprietary model. Due to the verification element, our system can benefit directly from additional test-time computation. Beyond numerical improvements, a key contribution is the verifiable reasoning process itself, offering interpretability not available from black-box models.

We were happy to discover that our development work using the Gemini-Flash LLM as a formaliser was directly transferable to a the open-licensed `Gemma2-it` model for the same role, with little loss of performance, enabling the whole pipeline to be run locally. The weakest link in the chain was, predictably, getting the 'Aha' of `wordplay` creation to work - humans can still generate wordplay that is beyond the capabilities of current models.

The authors sincerely hope that this work sparks interest in the cryptic crossword domain, which presents an array of interesting and challenging reasoning problems on which fruitful research can take place, even for those with limited computation budgets.

### 5.1. Further Work

Although we haven't explicitly tested generalizability to other NLP tasks, we believe the developed techniques – particularly formal verification and iterative refinement – offer valuable insights for improving LLM reasoning in complex NLU scenarios. Future research could fruitfully explore applying these techniques to other reasoning-intensive NLP tasks.

Around the time of the initial submission of this work, Guo et al. (2025) publicly disclosed a practical framework for learning to reason using Reinforcement Learning with an outcome-only reward scheme, which opens up a whole new avenue for investigation. While the OpenAI `o1` models had previously displayed reasoning traces, these were not considered in Table 1 : Partly due to cost/API considerations, but also because the authors strongly feel that proprietary black-box methods have limited research value.

Going forward, clearly a Reinforcement Learning approach would be very interesting to apply to the Cryptic Crossword domain, since these NLP reasoning problems are rather different from the mathematical proof / programming challenge tasks that are typical being tackled. Section 4.6 highlights a potential issue with our verification approach when combined with RL, since there a clear opportunity for RL reward hacking unless the verifier is made 'bullet-proof'.

We look forward to exploring the Cryptic Crossword reasoning task - there are a wide number of different avenues available.

## Acknowledgements

Support for this research was provided by the Google AI Developer Programs team, including access to the Gemini models and GPUs on Google Cloud Platform.

The authors thank the ICML reviewers for their time and valuable feedback.

## Impact Statement

### Societal impact

There are many current cryptic crossword enthusiasts that would potentially not welcome AI-enabled solvers to 'take over' their favourite pastime. In particular, when taken further, this line of work would potentially disruptive to public leaderboards that rank people according to the time taken to solve puzzles 100% correctly.

More generally, this paper presents work whose goal is to advance the field of Machine Learning. The techniques developed in this work extend work done in other domains for machine reasoning to a broader field that includes NLP/NLU tasks. That being said, we do not feel that there are significant societal consequences of our work that require specific highlighting.

### Potential bias in favour of native English speakers

While the English language has a high capacity for ambiguity and wordplay overall, making this type of crossword possible, cryptic crosswords also exist in other languages (Wikipedia contributors, 2024). In addition, although solving cryptic crossword answers may be very difficult (even for native English speakers), understanding the `answer` from given `wordplay` is much simpler.

## Reproducibility

### Datasets and Code

The following outline the efforts that have been made to ensure reproducibility:

- **Datasets** - Resources such as dictionaries used, and the Cryptonite and Wordplay datasets are available online, via the sources referenced in the main text.

- **System Code & LLM Prompts** - Python code for the complete end-to-end system is available under an Apache 2 license at `https://github.com/mdda/cryptic-crossword-reasoning-verifier`. The prompting required for the LLM formalisation are also included in the Appendix.

- **Models used / training** - The models referenced in this work are available with open weights (the Gemma2 models), or via API (the Gemini-Flash model, with a pinned version number). The training procedures are outlined in the text, and therefore have some degree of reproducibility. The fine-tuned Gemma2 models will be made available in any case, on publication.

### Computational requirements

The Gemini LLM was accessed by API, and the total spend to create the results in this paper was less than $100 USD, with each prompt round-trip taking around 5 seconds. The Fine-Tuning of the Gemma2 9B model took around 24 hours for a full Cryptonite training run, and 8 hours for the Wordplay dataset runs. Thus, the single-GPU model runs totalled less than $50 USD.

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

# A. Appendix

## A.1. Cryptic Crossword Background

The following borrows extensively from the description on Wikipedia (2024) (kudos to the authors there), to which we have added `wordplay` annotations in a notation typical of the `FifteenSquare.com` website (and in the Wordplay dataset use in this work).

### A.1.1. BASICS

A cryptic clue leads to its answer only if it is read in the right way. What the clue appears to say when read normally (the surface reading) is usually a distraction with nothing to do with the solution. The challenge is to find the way of reading the clue that leads to the solution.

A typical clue consists of two parts:

- The straight or definition. This is in essence the same as any non-cryptic crossword clue: a synonym for the answer. It usually exactly matches the part of speech, tense, and number of the answer, and usually appears at the start or end of a clue. For our annotations, the span that encompasses the `definition` is highlighted using curly braces.

- The cryptic, subsidiary indication or wordplay. This gives the solver some instructions on how to get to the answer in another (less literal) way. The wordplay parts of clues can be obscure, especially to a newcomer, but they tend to utilise standard rules and conventions which become more familiar with practice.

Sometimes the two parts of the clue are joined with a link word or phrase such as 'from', 'gives' or 'could be'. One of the tasks of the solver is to find the boundary between the definition and the wordplay, and insert a mental pause there when reading the clue cryptically.

We list below several of the important styles of `wordplay` that are commonly used, each with an annotated example. For a more comprehensive list, along with an outline of the 'Ximenean principles', please see Wikipedia (2024).

### A.1.2. ANAGRAMS

An anagram is a rearrangement of a certain section of the clue to form the answer. This is usually indicated by a codeword which indicates change, movement, breakage or something otherwise amiss. For example:

```
clue:       Chaperone shredded corset (6)
definition: {Chaperone} shredded corset
answer:     ESCORT
wordplay:   (corset)* (*shredded)
```

### A.1.3. CHARADE

In a charade, the answer is formed by joining individually clued words to make a larger word (namely, the answer). For example:

```
clue:       Outlaw leader managing money (7)
definition: Outlaw leader {managing money}
answer:     BANKING
wordplay:   BAN (outlaw) + KING (leader)
```

### A.1.4. CONTAINERS

A container or insertion clue puts one set of letters inside another. For example (also starting to add a little more indirection):

```
clue:       Utter nothing when there's wickedness about (5)
definition: {utter} nothing when there's wickedness about
answer:     VOICE
wordplay:   O (nothing) with VICE (wickedness) around it (about)
```

### A.1.5. DELETIONS

Deletion is a wordplay mechanism which removes some letters of a word to create a shorter word. For example:

```
clue:       Bird is cowardly, about to fly away (5)
definition: {Bird} is cowardly, about to fly away
answer:     RAVEN
wordplay:   [c]RAVEN (cowardly) – 'C' (i.e. circa, about) (–fly away)
```

### A.1.6. DOUBLE DEFINITION

A clue may, rather than having a definition part and a wordplay part, have two definition parts. For example:

```
clue:       Not seeing window covering (5)
definition: {Not seeing} {window covering}
answer:     BLIND
wordplay:   Double Definition (DD)
```

### A.1.7. HIDDEN WORDS

With hidden word clues, the solution itself is written within the clue – either as part of a longer word or across more than one word. For example:

```
clue:       Found ermine, deer hides damaged (10)
definition: Found ermine, deer hides {damaged}
answer:     UNDERMINED
wordplay:   [fo]UND ERMINE D[eer] (hides)
```

### A.1.8. HOMOPHONES

Homophones are words that sound the same but have different meanings, such as 'night' and 'knight'. Homophone clues always have an indicator word or phrase that has to do with being spoken or heard. For example:

```
clue:       We hear twins shave (4)
definition: We hear twins {shave}
answer:     PARE
wordplay:   "pair" (twins, "we hear")
```

### A.1.9. REVERSALS

A word that gets turned around to make another is a reversal. For example:

```
clue:       Returned beer fit for a king (5)
definition: Returned beer {fit for a king}
answer:     REGAL
wordplay:   (LAGER)< (beer, <returned)
```

## A.2. Wordplay Dataset

The Wordplay Dataset used in this work is extracted from websites where cryptic crossword enthusiasts post solutions to the puzzles published in major publications. Each completed puzzle is annotated by an solver who provides the community with `definition`, `wordplay` and `answer` fields for each of the approximately 30 clues in that day's grid.

For UK papers, these enthusiast websites include:

- [timesforthetimes.co.uk](timesforthetimes.co.uk) - Times, Times Quick

- [www.fifteensquared.net](www.fifteensquared.net) - Independent, Guardian, Financial Times

- [bigdave44.com](bigdave44.com) - Telegraph, Sunday Telegraph

The following is an example from the Wordplay dataset, formatted in YAML (the workings of this clue are illustrated in Figure 2c):

```
title: Financial Times 16,479 by FALCON
url: https://www.fifteensquared.net/2020/05/18/ \
     financial-times-16479-by-falcon/
author: teacow
clues:
- clue: '{Offer} of support also broadcast'
  pattern: '8'
  ad: D
  answer: PROPOSAL
  wordplay: PROP (support) + (ALSO)* (*broadcast)
- ...
```

In the above:

- `clue` is the original clue, as given to solvers, but with the 'regular crossword' `definition` portion highlighted with curly braces;

- `pattern` is the number of characters in the answer;

- `ad` (across/down) is potentially significant, because some clues include directional hints such as 'before' or 'upwards' which are only meaningful if the orientation of the answer within the grid is known;

- `answer` is the clue's final answer (not known to the solvers before solving); and

- `wordplay` is an informally annotated explanation of how the clue words act together to logically build the letters in the answer (the resulting grid letters typically being in upper case) - here the `*` symbol signifies that `ALSO` is to be anagrammed due to the anagram indicator (`broadcast`) in the clue.

The Wordplay dataset is publicly available as Andrews (2024). Note that care was taken to ensure that the training/validation/test splits follow those of the Cryptonite dataset (and the test set answers are deliberately scrubbed from the retrieved data by the provided scripts, to reduce the chance that they become training data for an over-eager crawling system).

## A.3. Choice of Cryptonite vs Rozner

At the start of this paper's research program, the Cryptonite dataset of Efrat et al. (2021) was chosen as being the focus, over the approximately contemporaneous dataset from Rozner et al. (2021) (denoted Rozner here), for the following reasons:

- Cryptonite was larger (523k clues, compared to 142k in Rozner)

- Cryptonite consists of clues from The Times and The Telegraph (whereas Rozner is the UK's Guardian). While these are all fine newspapers, it is clear that in the cryptic crossword community (found online via websites for wordplay discussions, or YouTube channels) that The Times is considered the Gold Standard of cryptic crosswords.

- Indeed, Connor (2024) - one of the Guardian's own cryptic blog posts - directly states: "The Times hosts an annual crossword-solving competition and it remains, until such time as the Guardian has its own version, the gold standard."
  - In the authors' view, The Times deserves its role as Gold Standard due to (a) adhering to / upholding the Ximenean standard Macnutt (1966) for what is allowed in clues; (b) doing so for decades; and (c) maintaining high consistency of clue difficulty within puzzles

- The Cryptonite dataset was made available for direct download - even though the licensing is (politely) 'fuzzy', it remains a useable research dataset (and seems unlikely to be challenged by The Times, since it is not possible to reconstruct their full puzzles from the clues given as individual line-items, due to deduplication, for example)

  - The Rozner dataset required researchers to 'scrape their own data', likely because while the data was being retrieved from a public website, the data itself could reasonably be assumed to be copyrighted. This slight inconvenience had a useful impact (please see below)

- Unlike the Cryptonite dataset, the Rozner dataset does not include Across/Down markers for the clues - which makes some of the clues difficult to resolve (for instance `EXAMPLE` on the paper's first page can only be read correctly if one sees that it is a Down clue - which converts 'up' into a reversal indicator)

- The Cryptonite dataset also includes 'is_quick' annotations that show whether a clue was taken from a 'Quick Cryptic' crossword (these clues are typically easier, which enables a further degree of performance analysis).

- The Cryptonite dataset splits were set in stone. Rozner, though, had a series of splits (random, disjoint, and 'init'):

  - The 'random' split was clearly shown to be a poor way of separating train/test due to close overlaps
  - The 'disjoint' split is similar in spirit to the Cryptonite methodology
  - The 'Init' split had the additional twist that common prefixes would only be found in their own splits. This had a catchy intuition, although it's not clear from a cryptic cluing perspective whether this has much genuine basis. While there are some prefixes that are common (eg: `EX−` is easily clued by referring to divorce, etc), the impact seems overall marginal (particularly given the accuracy rate differences reported)

Our paper describes a system trained on Cryptonite clue/answer training data, and also (as a component) the Wordplay dataset (which abides by the Cryptonite splits too).

It would be possible to test our existing (Cryptonite trained) system on the Rozner 'Init' test set. However, while Saha et al. (2024) could have the flexibility to run tests on either dataset (since no training was performed), running our current model on the Rozner 'Init' test set would be clearly mis-aligned vis-a-vis the data split.

But there is also a structural reason against re-training the paper's system on the Rozner 'Init' split for (specifically) Wordplay. The Wordplay dataset generation process was guided by the principle of maintaining the Cryptonite splits, it would be a disaster if Rosner 'Init' *Wordplay* splits were to be made public. The reason: It is very likely that the Cryptonite *test* set has a large intersection with the Rozner 'Init' *training* set (and conversely). As seems evident from the baseline improvements shown above, OpenAI likely trains on the Cryptonite training set (as they are welcome to do). *However*, since (as of November 2024) Saha et al. (2024) appears to have released (or re-released) the 'Init' training set under an MIT license, a commercial vendor such as OpenAI would be quite within their rights to also train on that. Thus, commercial systems (against which reviewers are forcing academic papers to benchmark) will have been trained on the test sets (without commercial vendors explicitly 'cheating' - they will just be training on all the available training data).

In the authors' judgement, the *reasoning paths* that are being tested here through the cryptic crossword task are a prize cultural asset, generated over decades of human effort, and this should not be squandered. Hopefully, this explains the authors' decision to train on only the Cryptonite dataset : We don't want to encourage the gathering and distribution of cross-contaminating datasets, specifically Wordplay datasets.

### A.4. Fine-tuning prompt

The following is a verbatim training example used for the fine-tuning of the `Gemma2-9B-base` model:

```
### Instruction:
Cryptic clue wordplay generation : Given the clue and the answer, \
return expert definition and wordplay annotations

### Input:
clue: "musical and ballet, oddly, that can be avoided"
answer: EVITABLE ~ evitable

### Response:
definition: musical and ballet, oddly, {that can be avoided}
wordplay: EVITA (musical) + B[a]L[l]E[t] (ballet, odd letters)
```

### A.5. In-Context Learning Prompts for the Gemini LLM

The Gemini LLM is prompted in-context with the concatenation of the following sections:

- Cryptic Crossword overview

- Many-shot wordplay examples

- Declaration of 'external' Python functions

- 6-shot formalisation demonstration

- Actual problem statement (for continuation as a Python proof)

- *After a verification failure*: Error messages for the generated proof, with hints if available, and request to improve iteratively

The sections of the prompt are described more fully below, note that care was taken to ensure that the chosen terminology was use consistently throughout.

### A.5.1. CRYPTIC CROSSWORD PREAMBLE

The following is the rubric and `wordplay` preamble given to the Gemini LLM:

```
A Cryptic crossword question involves using the words in \
the given clue to yield an answer that matches the letter pattern.
The clue will provide a definition of the answer, as well \
as some 'wordplay' that can also be used to confirm the answer.
Expert question solvers write informal 'proofs' using a \
particular format.

For the definition, the original clue is annotated with \
'{}' to denote where the definition is to be found.
For the wordplay, the following conventions are loosely used:
* The answer is assembled from the letters in CAPS
* Words in brackets show the origins of letters in CAPS, \
often being synonyms, or short forms
* Action words are annotated as illustrated:
  + (ETO N)* (*mad = anagram-signifier) = TONE
  + (FO OR)< (<back = reversal-signifier) = ROOF
  + [re]USE (missing = removal-signifier) = USE
* DD is a shorthand for 'Double Definition'
```

### A.5.2. MANY-SHOT WORDPLAY EXAMPLES

Around 20 examples from the Wordplay dataset are included in the in-context prompt:

```
For example:
---
clue: "arrived with an artist, to get optical device (6)"
definition: arrived with an artist, to get {optical device}
answer: CAMERA
wordplay: CAME (arrived) + RA (artist, short form)
---
clue: ...
```

### A.5.3. EXTERNAL PYTHON DSL FUNCTIONS

Domain Specific Python functions are described in-context to the LLM, which appears able to use them without seeing their internal functionality. In fact, the actual implementation of the functions is more extensive than described, since calls to these functions also track 'near misses' which can be fed back as hints during the re-write process.

```
The task is to produce a formal proof using python code, \
where the docstring will also include an informal proof as an aid.
The following are functions that can be used in your output code:

Action=Enum('Action', 'ANAGRAM,REMOVE_FIRST,INITIALS,REMOVE_LAST,'+
                       'GOES_INSIDE,GOES_OUTSIDE,REVERSE,SUBSTRING,HOMOPHONE')
# External definitions
def is_synonym(phrase:str, test_synonym:str, pattern:str='') -> bool:
  # Determines whether 'test_synonym' is a reasonable synonym for 'phrase',
  # with letters optionally matching 'pattern'
def is_abbreviation(phrase:str, test_abbreviation:str) -> bool:
  # Determines whether 'test_abbreviation' is
  # a valid abbreviation or short form for 'phrase'
def action_type(phrase:str, action:Action) -> bool:
  # Determines whether 'phrase' might signify the given 'action'
def is_anagram(letters:str, word:str) -> bool:
  # Determines whether 'word' can be formed from 'letters' (i.e. an anagram)
def is_homophone(phrase:str, test_homophone:str) -> bool:
  # Determines whether 'test_homophone' sounds like 'phrase'
```

### A.5.4. FEW-SHOT FORMALISATION EXAMPLES

The following are 3 (out of 6) of the few-shot formalisation examples given before the final test-case prompt:

````
The following are examples of simple functions that prove that \
each puzzle solution is correct:

```python
def proof(answer="ONCE",
          clue="head decapitated long ago", pattern='4'):
  """
  definition: head decapitated {long ago}
  wordplay: [b]ONCE (head decapitated = remove first letter of BONCE)
  """
  assert is_synonym("head", "BONCE")
  assert action_type("decapitated", Action.REMOVE_FIRST) \
          and "BONCE"[1:]=="ONCE"
  assert is_synonym("long ago", "ONCE", pattern='4')
proof()
```

```python
def proof(answer="DECIMAL",
          clue="the point of medical treatment", pattern='7'):
  """
````

```
  definition: {the point} of medical treatment
  wordplay: (MEDICAL)* (*treatment = anagram)
  """
  assert is_synonym("the point", "DECIMAL", pattern='7')
  assert action_type("treatment", Action.ANAGRAM)
  assert is_anagram("MEDICAL", "DECIMAL")
proof()
```

```python
def proof(answer="SUPERMARKET",
          clue="fat bags for every brand that's a big seller",
          pattern='11'):
  """
  definition: fat bags for every brand that's {a big seller}
  wordplay: SUET (fat) (bags = goes outside) of \
            (PER (for every) + MARK (brand))
  """
  assert is_synomym("fat", "SUET")
  assert action_type("bags", Action.IS_OUTSIDE)
  assert "SUET" == "SU" + "ET"
  assert is_abbreviation("for every", "PER")
  assert is_synomym("brand", "MARK")
  assert "SU"+"PER"+"MARK"+"ET" == "SUPERMARKET"
  assert is_synonym("a big seller", "SUPERMARKET", pattern='11')
proof()
```

### A.5.5. FORMALISATION INSTRUCTION

The following instruction is given before the final 'test-case' prompt illustrated in Figure 4:

```
# Please complete the following in a similar manner, and return the whole function:
```

```python
def proof(answer= ...
```

### A.5.6. PROOF VERIFICATION WITH HINTING

Examples of assertion failures, with constructive hinting, are shown:

```
AssertionError: assert: is_abbreviation('an Artist', 'RA') :
   'an Artist' does not have a valid abbreviation;
   'RA' is an abbreviation for : artist, artillery, Royal Artillery,
   gunners, painter
AssertionError: assert action_type('goes crazy', Action.ANAGRAM) :
  'goes crazy' itself does not suggest Action.ANAGRAM, but 'crazy' does
AssertionError: assert action_type('worked', Action.HOMOPHONE) :
  'worked' does not suggest Action.HOMOPHONE, but maybe Action.ANAGRAM

# Please re-implement the SOLUTION above \
(altering both the docstring and the python code as required), \
taking care to fix each of the problems identified, \
and return the whole function:
```

```python
def proof(answer= ...
```

Once the prover has fully parsed a given output with zero assertion failures, the proof is considered a success (up to 2 re-write iterations are allowed, more that that is considered an overall failure to prove the answer).

*Table 2.* Partial Correctness Metrics Results

| Model | known% | samples | Validation | | | Test | | |
|---|---|---|---|---|---|---|---|---|
| | | | Overall | Quick | Hard | Overall | Quick | Hard |
| GPT-4T ('Init') | 25% | | | | | 33.7% | | |
| GPT-4T ('Init') | 50% | | | | | 52.9% | | |
| GPT-4T ('Init') | 70% | | | | | 76.3% | | |
| Gemini-Flash | 25% | 200 | 37.0% | 38.5% | 36.9% | **45.5%** | 66.7% | 43.8% |
| Gemma2-9B-it | 25% | 200 | 37.5% | 38.5% | 37.4% | 44.0% | 66.7% | 42.2% |
| FastText k=1 NN | 25% | 200 | 15.5% | 15.4% | 15.5% | 21.0% | 33.3% | 20.0% |
| FastText k=1 NN | 50% | 200 | 52.5% | 38.5% | 53.5% | **62.0%** | 46.7% | 63.2% |
| FastText k=1 NN | 70% | 200 | 79.0% | 61.5% | 80.2% | **81.0%** | 100.0% | 79.5% |

### A.5.7. PARTIAL CORRECTNESS METRICS RESULTS

Our results in Table 2, which corresponds to the Exploiting Partially Filled Grids section in Saha et al. (2024), are based on running models on Cryptonite splits, rather than their 'Init'. However, the percentage differences observed are likely large enough outweigh the shift between the two datasets.

The GPT-4T rows in Table 2 are as reported in Saha et al. (2024), and apply to the 'Init' test dataset (i.e. different from our Cryptonite numbers, but still comparable figures in terms of what is being shown here).

The Gemini/Gemma2 rows show the effect of simply filtering the output of our Gemma2-9B fine-tuned candidate answer proposal model, based on a random letter pattern using the same formulation as Saha et al. (2024) and then using the rest of our pipeline. Here, our approach beats the previously reported GPT-4T results, and is itself limited by our first stage Gemma2 model's 'Top-20' candidate answers only containing the correct answer only around 45% of the time.

Our FastText $k = 1$ kNN systematic approach is clearly very powerful - particularly considering that it does not involve any large model, merely a brute-force search. This only works because the number of known letters for these rows are so high - indeed the 70% level would not be allowed in crosswords that obey the Ximenean guidelines of Macnutt (1966).

If solving complete grids were the target of our research, we would certainly incorporate this kind of solution and overlay the reasoning component to choose from the short-list output (rather than just selecting the first entry, as here). Note that also the performance is bounded above, because the wordlist is not exhaustive - we determined that 7.0% of the gold answers (on the Cryptonite test set) do not appear in the list. This may not be such an issue with the 'Init' dataset, since that wordlist is likely more restricted.

