# OpenReview forum: "A Reasoning-Based Approach to Cryptic Crossword Clue Solving"
_ICML.cc/2025/Conference — ICML 2025 poster_

### Official Review · Reviewer_9S8J · 2025-03-13

**Overall Recommendation:** 2

**Summary:**

This paper presents a multi-stage pipeline approach to solving cryptic crossword clues, focusing on reasoning through the wordplay mechanisms that make these puzzles challenging. The system consists of:

1. A candidate answer generator (fine-tuned Gemma2-9B model)
2. A wordplay suggestion generator (fine-tuned Gemma2-9B model)
3. A formalizer that converts wordplay into verifiable Python code (using either Gemini-Flash or Gemma2-9B)
4. A purpose-built verifier that evaluates the Python code and provides feedback for refinement

The key methodological contribution lies in the design of the Formaliser and verification system. This framework systematically decomposes model-generated solutions into verifiable sub-tasks through Python scripts, where each verification component either leverages LLM reasoning (via the `is_synonym` and `is_homophone_functions` functions) or employs traditional retrieval methods (via the `action_type` and `is_abbreviation` functions). This verification-based approach enables interpretable reasoning and provides a mechanism for iterative improvement of solutions.

The authors evaluate their system on the Cryptonite benchmark dataset and achieve 32.5% Top-1 accuracy with their Gemini-Flash Formaliser, outperforming GPT-4o (27.6%) on the same test samples. Additionally, they demonstrate that open-source models can achieve competitive performance (29.5%) when used throughout the pipeline.

**Claims And Evidence:**

- The paper's claim about providing verifiable reasoning procedure for cryptic clue solutions is well-supported through examples and is a clear strength of the approach.
- The paper's primary claim of achieving state-of-the-art results on the Cryptonite dataset is supported by experimental evidence, but the claimed improvements over GPT-4o (32.5% vs 27.6%) may be within the margin of error (stated as ±3.3% at 200 samples).
- While the experimental results support the conclusion that fine-tuned open-source models with less parameters can achieve competitive performance, the contribution of the Python formalization and verification components is less clearly established. The improvement from the Formaliser over the fine-tuned models is not particularly substantial and may fall within the margin of error, failing to conclusively validate the additional mechanisms' contribution beyond fine-tuning.

**Essential References Not Discussed:**

While the paper effectively connects to literature on mathematical reasoning and code verification, there's a potential connection to tool-integrated reasoning frameworks that isn't discussed. The paper frames its approach primarily in relation to Draft, Sketch, and Prove frameworks, but the methodology of using Python as an external verification mechanism for natural language reasoning problems also shares conceptual similarities with tool-integrated reasoning approaches. For example, the recent works in mathematical problem solving fields like [1] present many framework where language models leverage external tools to enhance reasoning capabilities.

[1] Gou, Z., et al. (2024). ToRA: A Tool-Integrated Reasoning Agent for Mathematical Problem Solving. https://arxiv.org/abs/2309.17452

**Experimental Designs Or Analyses:**

The paper fails to provide comprehensive timing and computational cost metrics. The formalization process substantially increases the pipeline's complexity, requiring additional model calls, verification steps, and potential rewrites. Without detailed per-step and end-to-end timing data, it is impossible to fairly compare this approach against simpler, more direct baselines. While the authors mention total costs under $100 USD, they do not break down the inference time or computational requirements at each stage and each inference experiment of their pipeline, making it difficult to assess the practical efficiency trade-offs between accuracy gains and increased computational overhead.

**Methods And Evaluation Criteria:**

### Methods:
The paper's method (especially the formalisation and verification parts) is derived from observation and emulation of human solver processes for cryptic crossword clue solving. By explicitly modeling how humans parse, reason through, and verify cryptic clues, the authors have designed a specialized verification workflow tailored to this specific task. The systematically decomposition aligns well with the inherent structure of cryptic puzzle solving.

### Evaluation Criteria:
The basic metrics are appropriate for the task at hand:
- Using the Cryptonite dataset is justified as it represents a standard benchmark in the field
- The Top-1 exact match accuracy is an appropriate primary metric for cryptic crossword solving
- The "Partial Correctness Metrics" in the appendix provide additional insights into system performance with varying levels of letter hints

**Other Comments Or Suggestions:**

No other comments.

**Other Strengths And Weaknesses:**

### Strengths:
- The paper provides good coverage of cryptic crossword conventions and mechanisms
- The step-by-step formalization of cryptic reasoning and the feedback mechanism for refining formalized proofs is well-designed

### Weaknesses:
- (Minor) Testing is limited to a single dataset (Cryptonite) rather than exploring transferability to other crossword sources

**Questions For Authors:**

No other questions.

**Relation To Broader Scientific Literature:**

Pros: The approach draws on the Draft, Sketch, and Prove frameworks, taking a decomposition-then-formal verification approach. The analogy is apt, and the paper provides valuable insights into designing verification functions without a formal DSL for verification. To some extent, it is inspiring for addressing key challenges in current AI research in the code and math domain, especially in developing O1-style reinforcement learning training methods where verification is critical.

Cons: However, current implementations are limited to relatively restricted task domains. Whether this approach can be generalized to more complex scenarios—especially those where human behavior is not easily observable or where specialized tools and sufficiently powerful verification models are not available—remains an open question. The broader applicability of this verification-centric approach to less structured reasoning tasks warrants further exploration.

**Theoretical Claims:**

The paper makes no theoretical claims.

---

> ### Author Rebuttal · Authors · 2025-04-01
>
> ### Statistical Significance of Improvement over GPT-4o
>
> We acknowledge the reviewer's point regarding the margin of error.  Since the improvement over GPT-4o on our sampled test set was within the simple margin of error, we performed the Bayesian IRT analysis presented in the paper that suggests a high probability (92%) that Gemini-Flash Formaliser is indeed better than GPT-4o. Beyond numerical improvements, a key contribution is the verifiable reasoning process itself, offering interpretability not available in black-box models.
>
>
> ### Contribution of Formalization/Verification Components
>
> We agree that demonstrating the isolated numerical contribution of the formalization and verification steps is challenging.  However, the ablation studies (Table 1 - AB lines) clearly show that removing these components significantly degrades performance, indicating they are not merely incremental but essential for the system's success.  These steps provide crucial interpretability and enable systematic error correction through feedback, which fine-tuning alone cannot achieve.  The focus is not solely on maximizing Top-1 accuracy, but on reasoned solutions.
>
>
> ### Lack of Timing/Cost Metrics
>
> We appreciate the reviewer's point about timing and computational cost.  Quantitatively comparing proprietary API-based models (like GPT-4o) with local open-source models on FLOPs is problematic due to the unknown parameter count/architecture of the proprietary models.  Moreover, the proprietary models are likely run on very capable hardware - so wall-clock timing comparisons also do not make much sense.  We mention the total cost was under $100 USD to highlight the feasibility and accessibility of our approach, especially using open-source models.  That being said, our research prioritized reasoning and interpretability over raw speed : Generating multiple candidate answers, followed by multiple wordplay samples, can been framed as inference-time computation (using 9B models) rather than using a large proprietary model.
>
>
> ### Limited Task Domain
>
> We recognize the domain-specificity of cryptic crossword solving.  However, we argue that it serves as a rigorous and complex testbed for reasoning, requiring multi-faceted language understanding and logic.  While the specific DSL is tailored to crosswords, the underlying principles of our approach – decomposition, formalization, and verification with feedback – are intended to be more broadly applicable to other reasoning tasks.  Cryptic crosswords, with their clearly defined rules and solutions, allow for precise evaluation and iterative refinement of these principles.
>
>
> ### Missing Reference (ToRA)
>
> We thank the reviewer for pointing out the relevance of tool-integrated reasoning frameworks and specifically for suggesting the ToRA paper [Gou et al., 2024].  We agree that our work shares conceptual similarities with this approach, and we will add a discussion of ToRA and cite this paper in the related work section.  A key novelty in our approach is the use of Python itself as the "tool" for verification within an NLP task, enabling a flexible and interpretable verification process without relying on pre-defined formal DSLs for the entire reasoning chain.
>
>
> ### Single Dataset Testing (Cryptonite)
>
> We acknowledge that testing is primarily on Cryptonite.  Appendix A.3 details the rationale for choosing Cryptonite over the Guardian dataset [Rozner et al., 2021], including dataset size; source ('gold standard' Times/Telegraph clues); and the consistent use of Cryptonite's train/val/test splits for focusing on reasoning vs the Wordplay Dataset.
>
>
> We believe our rebuttal addresses the insightful points raised in your review and provides important clarifications regarding the statistical significance of our results, the contribution of the formalization and verification components, the practical considerations of timing and cost, the domain-specificity of our approach, and the connection to tool-integrated reasoning frameworks. We would be grateful if you would consider re-evaluating your rating in light of these responses.  Thank you again for your time and constructive feedback, which has been invaluable in improving our work.

---

### Official Review · Reviewer_FiS7 · 2025-03-14

**Overall Recommendation:** 3

**Summary:**

This paper proposes a reasoning-based approach to solving cryptic crossword puzzles, integrating large language models (LLMs) with Python formal verification.
The system generates answer candidates, derives wordplay explanations, and translates them into verifiable Python code for validation.
It achieves a new state-of-the-art (SOTA) performance on the Cryptonite dataset.

**Claims And Evidence:**

Yes

**Essential References Not Discussed:**

None

**Experimental Designs Or Analyses:**

Yes

**Methods And Evaluation Criteria:**

Yes

**Other Comments Or Suggestions:**

None

**Other Strengths And Weaknesses:**

Strengths:
- The approach introduces Python formal verification, ensuring that each solution's reasoning process is transparent and verifiable. By converting wordplay explanations into executable Python code, the system eliminates the "black-box" problem of traditional LLMs.
- The study demonstrates that fine-tuned open-source models (Gemma2-9B) can outperform proprietary models like GPT-4o in complex reasoning tasks. This makes the solution cost-effective, locally deployable, and a strong alternative for NLP applications.

Weaknesses:
- The Python-based verification system has weaknesses, such as bypassing assert statements or generating logically inconsistent proofs. This reduces reliability and may lead to incorrect solutions being accepted.
- The approach is tailored for cryptic crossword solving, and its effectiveness in broader NLP reasoning tasks remains unproven. Additionally, the LLM struggles with highly complex or unconventional clues, limiting its generalizability.

**Questions For Authors:**

None

**Relation To Broader Scientific Literature:**

The paper  contributes to the intersection of natural language understanding (NLU), reasoning, and programmatic verification within the domain of cryptic crossword solving.

**Theoretical Claims:**

No Theoretical Claims

---

> ### Author Rebuttal · Authors · 2025-04-01
>
> We appreciate the reviewer highlighting the identified limitations of our Python-based verification system. To clarify, these shortcomings were not overlooked, but rather explicitly discussed in Section 4.5 ("Known Limitations of the System") to ensure transparency.  Presenting these potential 'shortcuts' was a deliberate choice to enhance the paper's rigor and openness, crucial for fostering further research – especially as it may serve as a cautionary note regarding the potential for a Reinforcement Learning loop (in future work).
>
> As to the reviewer's point about the challenges posed by complex and unconventional cryptic clues, while the paper doesn't explicitly claim that the LLM struggles with these in a way that fundamentally limits generalizability, we acknowledge that cryptic crosswords, by their very nature, present a spectrum of complexity, and some clues are indeed more challenging than others.
>
> Although we haven't explicitly tested generalizability to all broader NLP tasks, we believe the developed techniques – particularly formal verification and iterative refinement – offer valuable insights for improving LLM reasoning in complex NLU scenarios.  Future research could fruitfully explore applying these techniques to other reasoning-intensive NLP tasks.
>
> We sincerely appreciate the reviewer's valuable feedback, which has allowed us to clarify these key points. Having addressed the verifier limitations and the scope of generalizability, we respectfully request that the reviewer re-evaluate the paper's rating based on this improved understanding of our work.

---

### Official Review · Reviewer_c1BM · 2025-03-14

**Overall Recommendation:** 3

**Summary:**

The paper presents a reasoning-based system for solving cryptic crossword clues using open-licensed LLMs. It follows a three-step pipeline: (1) generating answer candidates, (2) proposing wordplay explanations, and (3) verifying solutions via a Python-based formalizer. The system outperforms prior methods on the Cryptonite dataset, achieving a Top-1 accuracy of 32.5%, surpassing both rule-based and previous LLM-based approaches. The key contribution is using Python assertions to validate reasoning, improving reliability and interpretability.

**Claims And Evidence:**

Most claims are well-supported, particularly the state-of-the-art performance (32.5% Top-1 accuracy) on Cryptonite and the effectiveness of wordplay verification. However, the Python-based verifier has limitations (e.g., bypassing assertions), meaning correctness is not fully ensured. The claim of broader generalization is weak, as the study focuses only on cryptic crosswords. More analysis of failure cases and alternative tasks would strengthen the paper’s broader impact.

**Essential References Not Discussed:**

N/A

**Experimental Designs Or Analyses:**

Log probability-based ranking ablations suggest verification matters, but the study lacks detailed error analysis on failure cases.
The paper would benefit from qualitative examples of incorrect reasoning that passes verification.

The verifier can be bypassed via faulty assertions, but there is no analysis of how often verification fails or of its impact on final accuracy.

**Methods And Evaluation Criteria:**

Yes, the methods and evaluation criteria are well-suited for cryptic crossword solving. However, there is no evaluation of the boarder application of the proposed method.

**Other Comments Or Suggestions:**

N/A

**Other Strengths And Weaknesses:**

N/A

**Questions For Authors:**

N/A

**Relation To Broader Scientific Literature:**

The paper proposes an LLM-based and program-aided approach to significantly improve machine performance on the cryptic crossword solving task.

**Theoretical Claims:**

N/A

---

> ### Author Rebuttal · Authors · 2025-04-01
>
> In Section 4.5 ("Known Limitations of the System"), we chose to explicitly discuss the limitations of our Python-based verification system to ensure transparency.  However, these potential 'shortcuts' are mainly a cautionary note regarding the potential for a Reinforcement Learning loop (in future work), where we would expect an RL system to exploit them, whereas the current system only rarely showed such behaviour (which is why these loopholes were not eliminated during development).
>
> On the other hand, we would be delighted to add a section containing a qualitiative analysis of the key false positive / negative failure modes of the system.  Notably:
>
> * The headline success rate is bounded above by the initial candidate answer generation process.  If the system cannot guess the answer in its top-k (k=20 here), the remaining process is doomed.  As shown in Figure 7a, even with higher top-k, this puts an upper bound on performance that is well below 100% correct.  Having better candidate answer generation would be beneficial - and this would directly feed through our verification process (which is a step that the proprietary models do not benefit from, and gives us human-interpretable reasoning for each solution)
> * A significant source of false negatives is the `is_synonym` function, which relies on a sequence of steps: first we attempt a look-up in an open-source thesauraus, then in a dataset of 'regular crossword answers'.  But the final fall-back is asking an LLM whether given phrases are synonyms.  While the first two steps may vote positively (for easy matches), it is common in cryptic clues that the `definition` and the `answer` are more distantly related than regular crosswords.  For instance, in Appendix A.1.7, we have the true answer `UNDERMINED` being defined by `damaged`.  This would likely be too distant to be reasonable for a regular crossword, but the strength of the wordplay (the answer being literally given in the clue) is confirmation enough to satisfy solvers.  Setting this 'synonym distance hurdle' is an ongoing challenge.
>
> Clearly, this analysis deserves more space than was available in the submitted version of the paper, and would provide valuable additional insight to those interested in the issues being tackled.  Thank you for highlighting the need for this quantitative analysis.

---

### Official Review · Reviewer_HqW1 · 2025-03-17

**Overall Recommendation:** 4

**Summary:**

This paper presents a system for solving cryptic crossword clues using a collection of fine-tuned and ICL-prompted open-weight language models, as well as a custom Python-based domain-specific interpreter. The proposed system first samples a set of candidate solution words from a fine-tuned proposal model. Another fine-tuned model is then used to sample potential wordplay decompositions of the answer candidates. An ICL prompted model then translates these wordplay breakdowns into a set of Python assertions containing a mixture of vanilla expressions and special functions (e.g. `is_synonym(a, b)`, `is_homophone(a, b)`) backed by additional small LM calls. Two rounds of refinement are conducted on this translation based on interpreter feedback. The highest-frequency answer with a wordplay breakdown that passes its translated assertions is returned as the answer to the clue (or the highest-frequency answer if no wordplay passes verification).

The authors evaluate their method on the Cryptonite dataset, containing cryptic crossword clues from the Times and Telegraph UK newspapers. They compare their method to several single-stage few-shot prompted LMs, including GPT-4o and Gemini 1.5 Flash. On the Hard subset of the test split, their method outperforms all considered baselines.

**Claims And Evidence:**

The authors claim that their method achieves SOTA results on Cryptonite. Section 2.2.2 does mention that the lack of reasoning capabilities presented a problem for models prior to 2024, but makes no mention of the new generation of LLMs post-trained for reasoning (the only mention I could find is a namedrop on line 207), begging the question of how the proposed method would perform against DeepSeek R1, Gemini Flash Thinking, OpenAI O1 or Claude 3.7 Sonnet.

**Essential References Not Discussed:**

I think it might be worth discussing the connection to prior LM+autoformalization approaches - SatLM (Ye et al, 2023) comes to mind, especially since that work also deals with the LM language familiarity issue by using Python as an intermediate language to decode constraints from the model.

**Experimental Designs Or Analyses:**

The main comparison is sound. I also checked the evaluation setup for the partially-filled answer setting ($\S$3.6, A.3.5.7), and nothing stood out to me as improper.

**Methods And Evaluation Criteria:**

Yes, the methods and benchmark are appropriate.

**Other Comments Or Suggestions:**

No typos stood out to me during my reading. See "Other strengths and weaknesses" for a suggested change to Section 3.

**Other Strengths And Weaknesses:**

I thought the authors did a good job of motivating the problem area as a useful exercise, and in introducing the problem format, which is admittedly confounding at first glance.

Parts of the paper were a bit hard to follow or lacked important detail, namely the method description ($\S$3); while Fig. 1 shows a clear overview of the modeling flow, Section 3 does not contain a complete text description of the pipeline; it contains a stepwise summary of the human strategy that inspired the method, and subsections 3.1-3.5 detail stages of the process and design considerations for those stages, but there is no concise description of the overall procedure, which would be helpful to lead us through the subsections.

**Questions For Authors:**

One key question that I was unable to find in the text (although I may have missed it) is: How many wordplay/definition suggestions are generated per answer candidate? Parts of the text allude to checking multiple wordplay options, but I'm not sure if this is just due to considering multiple answer candidates.

**Relation To Broader Scientific Literature:**

The authors cite the prior rule-based SOTA system for cryptic crossword solving. Their general scheme of checking "loose" answer proposals from an LM has gained a lot of steam recently in the guise of "reasoning models", LMs post-trained with CoT+RL which can exhibit self-verification behavior. So far, public reasoning models have not been developed that are trained end-to-end to take advantage of autoformalization; such a direction is very promising. This work definitely serves as positive evidence of the power of this kind of architecture, even using small models and a semi-informal format.

**Theoretical Claims:**

N/A

---

> ### Author Rebuttal · Authors · 2025-04-01
>
> ### Answer to "How many wordplay/definition suggestions are generated per answer candidate?"
>
> For each clue, we generate 20 candidate answers (the cumulative probability graph for the upper bound of success after this is given in Figure 7a).  These are then deduplicated, and we create 5 definition and wordplay examples for each candidate answer (we previously experimented with 10 samples, but the difference was marginal).  Examples of wordplay for correct and incorrect candidate answers are given in Section 4.2.  From a human point of view, the wordplay explanations for incorrect candidate answers are clearly nonsense - but (as commonly seen using LLMs) the models tend to 'approve' of their own outputs.  Therefore, proving out the reasoning in a more concrete way is essential, so we use our novel 'formalization/verification' process on all of these outputs.
>
> Thank you for the question : We will certainly add this important experimental detail to the paper, as well as expanding the process flow explanation in the text of Section 3 (rather relying so heavily on Figure 1).

---

### Decision · Program_Chairs · 2025-05-01

**Decision:**

Accept (poster)

**Comment:**

This paper argues for the use of UK-style cryptic crossword clues as a source of test material for reasoning in AI models.
The case is well made:
- the volume of examples is considerably higher than other sources e.g. math olympiad, and new examples are produced daily by major news sources;
- the type of reasoning required is somewhat orthogonal to mathematics, but is certainly considered to require human-level skills
- the agentic verifier introduced is interesting, generating verifiable Python code.

Criticisms:
- Restricted to one source of examples.  The paper and appendix argue in terms of dataset leakage and dataset size that only the Cryptonite benchmark (520K examples) should be used, and not the Guardian one (140K); but many of these arguments apply more to the fine-tuning scenario, and could be otherwise overcome.  The argument is lyrical ("reluctance to dataset-hop"), but again this is refuted by paper's observation that new examples are produced at the rate of perhaps thousands per year.  If number of examples is a metric, then the union of datasets is larger than any single one.
It is notable that the appendix observes that "the Guardian clues can often be rather haphazard".  It is clear that such haphazard clues will be less suitable for the python verifier in this paper, so that scores would be expected to be lower.  The other arguments are valid, but the relative difficulty should be addressed more directly than as one sentence in a page of reasoning.  The absence of across/down annotation is again simply a contributor to difficulty, not a fundamental stumbling block.

- Comparison to reasoning models.  Models such as OpenAI 01 are dismissed in lines 206-214, without any evidence of an attempt to empirically test them.  When brought up by a reviewer, this point was ignored.  This is an important comparison - even if those models are better than the proposed one, the proposed work is valuable - but ignoring them or trying to argue them away weakens this paper.

A non-criticism is the failure to provide cost metrics - as the authors present the dollar cost of less than $100, this is sufficient to argue that as a research activity, this is not cost-prohibitive.  As emphasized by the authors, the objective is to discover frontiers of reasoning ability more than to create a low-cost system to solve crossword clues.